# CONCEPT-AWARE BATCH SAMPLING IMPROVES LANGUAGE-IMAGE PRETRAINING

## ABSTRACT

What data should a CLIP model see? Many data curation efforts aiming to answer this question center on the *quality* of a dataset. However, recent work has shown that while admitting impressive performance benefits, none of these curation methods are *concept-centric*, leading to them inheriting the biased properties of web-scale data distributions. In this work, we go beyond such concept-agnostic methods and advocate a more flexible *online concept-based curation* approach. To enable this, our first contribution is DATACONCEPT, a collection of 128M web-crawled image-text pairs annotated with fine-grained details about their concept composition. Building on DATACONCEPT, we fill another critical gap in the literature: the lack of a competitive, open-source alternative to highly performant batch sampling methods for Language-Image Pretraining. Specifically, we introduce **C**oncept-**A**ware **B**atch **S**ampling (CABS), a simple yet effective batch-sampling algorithm that distills batches with the broadest set of available concepts. Through rigorous evaluation on a broad suite of 28 benchmarks, we demonstrate that CABS significantly benefits Language-Image Pretraining (LIP) and yields highly performant models on long-tailed evaluations (up to +2.4 p.p. on Let-it-Wag!), while enabling practitioners to define custom concept distributions that optimize for specific downstream tasks. Importantly, with only one hyperparameter tuned for a single (backbone, eval) combination only, CABS shows full compatibility with both CLIP and SigLIP models. Both DATACONCEPT and the source code for CABS will be released.

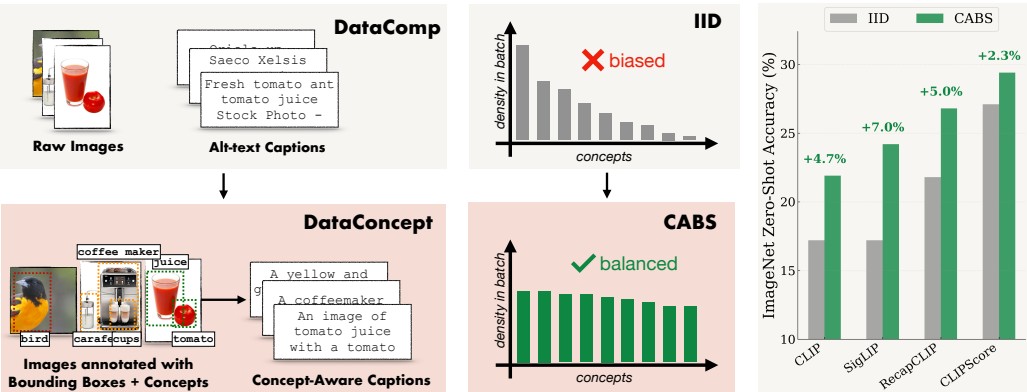

Figure 1: **Our two main contributions: DATACONCEPT and CABS**: We introduce a new large-scale concept-annotated dataset that enables a new form of concept-aware batch sampling, which improves vision-language pretraining.

## 1 INTRODUCTION

Web-scale pretraining datasets have been a critical ingredient in enabling the impressive generalisation of Vision-Language Models (VLMs). Consequently, in parallel to the release of state-of-the-art VLMs (Li et al., 2021; 2022), significant research and engineering efforts have been devoted to collecting and

open-sourcing such datasets (Sharma et al., 2018; Chen et al., 2015). The advent of CLIP (Radford et al., 2021), trained on a private collection of 400M image-text pairs, further motivated the open development of even larger, billion-scale datasets exemplified by LAION (Schuhmann et al., 2022) or DataComp-1.4B (Gadre et al., 2023). Although the size of these pretraining datasets is an influential aspect, their *quality*, as indicated by Nguyen et al. (2022); Gadre et al. (2023); Goyal et al. (2024), is equally as important, if not more. To improve quality, there exist various ways to *curate* a dataset, ranging from filtering according to well-defined metrics (*e.g.*, CLIPScore), to rewriting the crawled alt-text into a more detailed, meaningful description (Nguyen et al., 2023; Li et al., 2024).

Recently, Udandarao et al. (2024); Wen et al. (2024); Parashar et al. (2024) found that web-scale datasets exhibit extremely long-tailed concept distributions, contributing to biased downstream performance. However, none of the widely adopted curation strategies (*e.g.*, those benchmarked by DataComp (Gadre et al., 2023)) address this issue. In other words, they are *concept-agnostic*. A notable exception, and perhaps most related to ours, is MetaCLIP (Xu et al., 2024), an *offline* curation strategy that yields a more balanced distribution over CLIP concepts through metadata selection.

In this work, however, we depart from the offline MetaCLIP paradigm and advocate for a much more flexible alternative: *online concept-based curation*. Our rationale is simple: there is no "universal" notion of quality, and importantly, as we show in our experiments, downstream evaluation might bias what the optimal concept distribution looks like (Mizrahi et al., 2025; Abbas et al., 2024b). Therefore, we aim to show that incorporating concept-level information *during* pretraining, without discarding any data *a priori*, provides a complementary and effective avenue for VLM data curation.

To achieve this goal, we introduce DATACONCEPT: an open-source multimodal pretraining dataset with 128M image-text pairs from DataComp Gadre et al. (2023), fully annotated with grounded concept information. In DATACONCEPT, each sample comes with ① semantic concepts, ② bounding boxes, ③ per-concept confidence scores, and ④ high-quality, concept-driven synthetic captions obtained from a state-of-the-art VLM (Wang et al., 2024). With DATACONCEPT, we can therefore ask: *How can we effectively leverage visual concepts during vision-language pretraining?*

Based on this premise, we introduce a new training paradigm: **C**oncept-**A**ware **B**atch-**S**ampling (CABS). In contrast to offline, static curation, we do *not* impose a fixed, predetermined concept distribution, but rather enable flexible control over *online concept-based batch creation*. One variant of CABS selects samples based on the *diversity* of their constituent concepts. This scheme is in line with the MetaCLIP approach and significantly benefits zero-shot classification, especially over *long-tailed* evaluations. Additionally, we show that CABS is *not* limited to such a variant, but allows for more flexible selection criteria tailored to different downstream tasks, such as image-text retrieval, which may benefit from a different concept distribution during training (Abbas et al., 2024b).

Importantly, while other highly performant instances of batch-sampling for language-image pretraining exist (Udandarao et al., 2025; Evans et al., 2024a), they are closed-source algorithms, making CABS a competitive open-source alternative. Taken together, our **contributions** are:

1. DATACONCEPT: a new, *concept-centric* pretraining dataset for VLMs comprising 128M samples, which augments samples with fine-grained concept annotations and concept-driven synthetic captions. This helps enable further exploration of concept-centric data curation algorithms, a relatively underexplored avenue.

2. CABS: a new paradigm for vision-language pretraining that involves *online data curation* through *concept-aware batch sampling*. Paired with DATACONCEPT, the CABS paradigm enables dynamically control over the concept distribution of the data used throughout training.

3. Extensive evaluation on 28 benchmarks, 4 visual backbones, and 2 training objectives (CLIP vs SigLIP), demonstrates that CABS is highly effective for vision-language pretraining, while offering complementary benefits to existing curation recipes. CABS also represents a strong, open-source alternative to proprietary batch-sampling algorithms.

We hope this work encourages further study into concept-awareness as a critical dimension of improving data quality for VLMs.

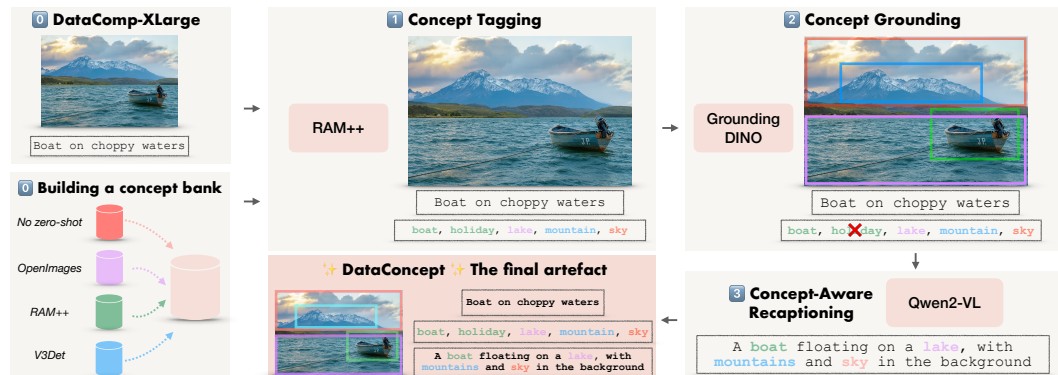

Figure 2: **DATACONCEPT.** We start by crawling images from DataComp (Gadre et al., 2023) and building a concept bank $\mathcal{C}$ by merging, deduplicating, and filtering various sources. Subsequently, ① *First-order tagging* assigns a preliminary list of concepts (from $\mathcal{C}$) to each sample; ② We *ground* each concept in the image, removing noise in initial candidates; lastly, ③ we rewrite alt-texts into *context-aware captions*.

## 2 DATACONCEPT: CONCEPT-AWARE DATASET AUGMENTATION

In this section, we introduce the pipeline used for constructing our large-scale, *concept-annotated* pool of 128M image-text pairs, DATACONCEPT. In the next section, we demonstrate the usefulness of our annotations by presenting the algorithm they enable and the resulting performance benefits.

**Initial pool.** We start with the DataComp's medium pool consisting of 128M image-text pairs (Gadre et al., 2023). The standard protocol for downloading the dataset suffers from significant link-rot.[1] Hence, we opt for randomly sampling a 128M subset from Datacomp's XLarge pool (12.8B).

**Building a concept bank.** The first step for annotating our pool is determining a *concept bank*, *i.e.*, the set of concepts that we seek to detect and tag. Previous work (Udandarao et al., 2024) curated a concept bank but it is rather limited ($4,029$) due to being constructed from 27 evaluation datasets. For broader coverage, we further source concepts from the class labels used in RAM++ (Huang et al., 2023), V3Det (Wang et al., 2023), and OpenImages (Kuznetsova et al., 2020), resulting in $19,261$ concepts, after de-duplication and safety removal. More details can be found in Appx. A.1.

**Concept tagging.** Equipped with an expansive concept bank, following Udandarao et al. (2024), we employ the RAM++ model to provide multiple concept tags for each sample in our pool.

**Concept grounding.** While RAM++ annotations provide fine-grained concept annotations per sample, we find that *(i)* RAM++ can be miscalibrated in its confidence predictions due to the extreme diversity of our concept bank, and *(ii)* RAM++ only provides a list of concept tags, without localising them in the image, which can lead to incorrect grounding. Thus, we use GroundingDINO (Liu et al., 2024) to additionally provide concept-specific bounding boxes. To enable precise localization of concepts, we propose two methods: *(i) Confidence seeding* – we feed RAM++ concept tags per sample (only those with at least $0.75$ confidence) as seed prompts to GroundingDINO – and *(ii) Resolution ensembling* – we use Weighted Box Fusion (Solovyev et al., 2021) to ensemble the GroundingDINO predictions over multiple image resolutions of $\{384, 512, 800, 1000\}$. These enable us to reduce hallucinations without significantly increasing processing latency. With the two aforementioned steps, we tag all samples in our pool using $6,201$ concepts, i.e. $\mathcal{C}$, the concept vocabulary for CABS.

**Concept-aware recaptioning.** Lastly, we augment each sample with a text caption by using a *concept-aware captioner*. Synthetic re-captions improve training data quality by reducing the noise in alt-text captions (Nguyen et al., 2023; Faghri et al., 2025; Fan et al., 2023). In our pipeline, we use Qwen2-VL-7B (Wang et al., 2024) to recaption each image in a *concept-aware manner*: we provide the list of detected concepts and the original caption in the prompt to the model for recaptioning.

---

[1]We successfully downloaded only 79% of the medium scale, as of 28/09/2024.

**DATACONCEPT**. Our multi-stage pipeline, fully summarised in Fig. 2, yields our final dataset. Each image-text sample in our dataset consists of concept metadata including coarse-grained tags with confidence scores, localised bounding-boxes, and concept-aware synthetic captions. In the next section, we describe how we can leverage these annotations to improve language-image pretraining.

## 3 CABS: CONCEPT-AWARE BATCH SAMPLING

Like all uncurated web-crawled datasets, DataComp pool exhibits natural frequency imbalances, leading to highly skewed concept distributions under IID sampling. These imbalances, in turn, bias batch-level supervision during pretraining. The core idea of the CABS paradigm is to counteract this bias by establishing a flexible target concept distribution over the data pool and selecting samples that approximate it. To make this possible, we leverage annotations from our proposed DATACONCEPT, where each image-text pair $(\mathcal{I}_i, \mathcal{T}_i)$ is complemented with a set of semantic concepts $\mathcal{S}_i$ from the concept vocabulary $\mathcal{C}$. This augmentation enriches the dataset with structured metadata, which in turn enables flexible batch construction protocols according to the downstream task of interest.

**Formulation**: We now formulate CABS as a sampling function. Given a filter ratio $f \in [0, 1)$ and a super-batch of size $B$, our objective is to extract sub-batches of size $b = (1 - f)B$ according to different heuristic functions, conditioned on the sample-level concepts $c$ as priors. The heuristic sampling function can flexibly be implemented to enable different batch compositions. For example, one heuristic variant can explicitly seek to be diversity-maximizing (like MetaCLIP) while another variant can seek to prioritize unique concepts. As we will show in Secs. 4 and 5, this inherent flexibility of our CABS paradigm can be leveraged to target different tasks of interest. Next, we develop a variant of CABS that explicitly aims to maximize concept diversity within a batch.

### 3.1 CABS WITH DIVERSITY MAXIMISATION (CABS-DM)

We introduce CABS-DM, desiged to mitigate the inherent long-tailed nature of concept distributions in image-text data. Our desired target batch is a near-uniform concept distribution, subject to lower and upper bounds on concept frequencies. This is denoted by $t_c$, the target count for concept $c$, measured by dividing budget $b$ evenly across the concept spread of $B$.

Since annotated samples are often multi-label (see Fig. 8), we denote the sample level concept set as $s_i \subseteq C$. CABS-DM proceeds by iteratively selecting the sample that maximises a gain function $g(i)$ and updating the sub-batch concept count $n_c$ (how many times concept $c$ has been selected) for all $c \in s_i$. This process continues until the desired batch size for training is obtained. The distilled sub-batch is vastly different from an IID-sampled batch, as illustrated in Fig. 3. An average CABS-DM sub-batch contains more than $1.5$ times the concepts of an IID-sampled batch and, in addition to exhibiting a flat concept distribution. This helps increase diversity and uniformity at the batch construction level.

CABS-DM includes the following components:

**Pooling Concepts and Target Count.** For each super-batch of size $B$, the global frequency $f_c$ of each concept is computed across several distributed processes. Additionally, we fix the (roughly uniform)

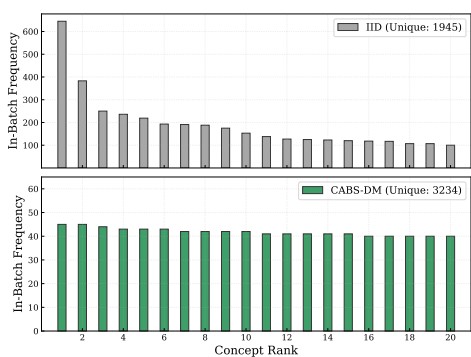

Figure 3: **Comparing sub-batch compositions.** While IID-sampling represents a strong skew in concept frequency, CABS-DM debiases the contrastive training objective with a near-uniform distribution.

target count $t_c$ for concept $c$, *i.e.* the maximum number of times $c$ should appear in the sub-batch, to enforce approximate uniformity. In a simplified setting, if each sample comprises 1 concept, $\sum_c t_c \approx b = (1 - f)B$, the sub-batch size.

**Gain Function.** Central to the effectiveness of CABS-DM is the sample-level gain or utility based on the current state of the sub-batch's concept distribution. Given super-batch sample $i$,

For concept set $s_i$ of sample $i$, define the gain as

$$g(i) = \frac{1}{|s_i|} \sum_{c \in s_i} \begin{cases} \dfrac{t_c - n_c}{t_c} + \dfrac{1}{f_c}, & \text{if } n_c < t_c, \\ -0.5, & \text{if } n_c \geq t_c \end{cases}$$

Each concept contributes a `balance gain`$(t_c - n_c/t_c)$ that encourages underrepresented concepts and a `long-tailedness bonus`$(1/f_c)$, when $n_c < t_c$ or a $-0.5$ penalty if $n_c$ reaches or exceeds $t_c$. At each step, we sort all remaining super-batch samples by this gain score and deterministically insert the top one into the sub-batch. CABS-DM then selects sample $i^\star = \arg\max_i g(i \mid n)$, appends $i^\star$ to the sub-batch, and updates $n_c \leftarrow n_c + 1$ for all $c \in s_{i^\star}$. If $c$ exceeds the maximum in-batch frequency, all remaining samples containing $c$ are invalidated.

CABS-DM *proceeds through a sequence of local maximisations to yield a balanced and diverse sub-batch*. At every iteration, it deterministically selects the sample with the highest gain, conditioned on the current sub-batch composition, without randomness. The deterministic nature of CABS-DM ensures reproducibility across runs for the same super-batch. The dual gain terms promotes balanced batch composition and concept coverage, directly addressing the biases of an IID-sampled batch.

## 4 EXPERIMENTS

We present an extensive evaluation of CABS-DM—in Sec. 4.2, we show that CABS-DM significantly outperforms IID sampling for language–image pretraining. In Sec. 4.3, we compare CABS-DM against open-source, state-of-the-art batch-sampling methods for Vision-Language Pretraining. Finally, in Sec. 4.4, we dig into CABS-DM's performance when training is *data-* vs *compute-constrained*, showing full compatibility with SOTA data filtering methods (*i.e.*, CLIPScore).

### 4.1 SETUP

**Models.** We train a ViT-B-32 (Dosovitskiy et al., 2020) CLIP using 224 image-resolution and ViT-B-16 SigLIP (Zhai et al., 2023) at 256 resolution. We further test CABS-DM by training ViT-S-16 CLIP and ViT-SO400M-14 SigLIP (Alabdulmohsin et al., 2023) models in Appx. B.

**Data.** For each model, we train two variants on DATACONCEPT: one using noisy alt-texts and another with our *concept-aware re-captions*. Note that IID sampling with alt-text corresponds to training on 128M samples from DataComp.

**Benchmarks.** Following Gadre et al. (2023); Abbas et al. (2024b); Udandarao et al. (2025) we consider a diverse pool of 25 classification and 2 image-text retrieval benchmarks, spanning fine-grained, object-centric, and scene-centric categories. Additionally, to assess the effectiveness of CABS-DM in the long-tail setting, we test on the "Let-It-Wag!" test set from Udandarao et al. (2024).

**Training.** We fix the training budget to 128M samples seen, which is equivalent to the *medium* scale of the DataComp benchmark. We closely follow the hyper-parameters set by DataComp for fair comparison across settings, including a batch-size of 4096. Unless otherwise specified, we set the filter ratio to $f=0.8$, thereby sampling from superbatches of size $B=20480$.

### 4.2 CABS-DM IMPROVES MULTIMODAL PRETRAINING

We comprehensively evaluate the effectiveness of CABS-DM against standard IID sampling for language–image pretraining. As shown in Tab. 1, CABS-DM consistently delivers improvements across four different evaluation settings. On ImageNet zero-shot classification, CABS-DM yields substantial gains over IID sampling, with an absolute improvement of $+5.0\%$ for CLIP ViT-B-32 and $+6.9\%$ for SigLIP B-16-256. Similar trends are observed across the broader suite of classification benchmarks, where CABS-DM provides higher average accuracy (final column of the table). Beyond standard benchmarks, CABS-DM also enhances long-tailed recognition: on Let-It-Wag!, we observe consistent boosts of $1.0-2.4\%$. This demonstrates that CABS-DM not only strengthens general representation learning but also better equips models to handle challenging, imbalanced distributions.

Importantly, we also observe consistent improvements in using our concept-aware re-captions vs alt-text, even with standard IID sampling. For example, with CLIP-ViT-B/32@224, our re-captions

Table 1: **CABS-DM improves over IID sampling for pretraining.** Our method substantially outperforms standard IID sampling, across settings. Importantly, gains from CABS-DM extend to the long-tailed "Let-It-Wag!" too, demonstrating the utility of our diversity-maximization procedure.

| Method | Captions | Zero-shot Classification | | | | Retrieval | | Let-it-Wag! | Avg (Clf) |
|---|---|---|---|---|---|---|---|---|---|
| | | IN-Val | IN-shift | Obj | Scene | COCO | Flickr | | |
| ViT-B-32-CLIP | | | | | | | | | |
| IID | alt | 17.3 | 15.2 | 32.3 | 36.4 | **9.7** | **16.2** | 5.1 | 28.2 |
| CABS-DM | alt | **21.9** | **18.6** | **34.2** | **37.0** | 7.7 | 12.6 | **7.5** | **31.0** |
| IID | recap | 21.7 | 20.8 | 36.4 | 43.1 | **24.0** | **41.3** | 5.9 | 33.0 |
| CABS-DM | recap | **26.7** | **25.4** | **39.6** | 42.8 | 22.8 | 34.4 | **7.1** | **35.5** |
| ViT-B-16-SigLIP-256 | | | | | | | | | |
| IID | alt | 17.2 | 15.3 | 29.6 | 35.9 | **11.1** | **18.9** | 5.2 | 26.4 |
| CABS-DM | alt | **24.1** | **20.8** | **33.5** | **39.6** | 10.1 | 14.1 | **7.0** | **30.9** |
| IID | recap | 28.8 | 27.4 | 41.5 | 48.9 | **37.1** | **57.0** | 6.6 | 38.6 |
| CABS-DM | recap | **34.7** | **32.3** | **43.2** | **50.6** | 36.0 | 51.4 | **7.6** | **41.1** |

contribute to a $+4.3\%$ improvement on ImageNet-1k, and $+4.8\%$ on average for zero-shot classification. For SigLIP-ViT-B/16@256, the same boosts are as large as $+11.6\%$ and $+12.2\%$.

**Remark on retrieval.** While performance on classification tasks is strong, CABS-DM seems to regress retrieval abilities. We posit why in Sec. 5, where we showcase the flexibility of the DATACONCEPT+CABS framework in controlling the concept distribution, and introduce a simple alternative (CABS-FM) that largely benefits retrieval. Note, however, that our concept-aware recaptioning provides substantial benefits for retrieval vs standard alt-text.

## 4.3 COMPARISON WITH ONLINE BATCH SAMPLING METHODS

In this section, we compare CABS-DM with GRIT-VLP and MAFA (Byun et al., 2022; 2024), two popular batch sampling methods. We remark that, although Evans et al. (2024a) and Udandarao et al. (2025) are also valid baselines, they are proprietary algorithms with no public implementation.

**Baselines.** Both GRIT-VLP and MAFA are sampling methods that rely on embedding similarity to sample hard-negatives. Despite their commonalities, they differ in a crucial detail: GRIT's hard-negatives are based on the *current state* of the learner, while MAFA relies on similarities computed by a *pretrained* model. In the original paper, MAFA uses BLIP to sample hard-negatives for the learner. However, BLIP is trained at a much smaller budget than current standards. To avoid this confounder in our comparison, we pretrain CLIP and SigLIP models with a compute budget of 128M samples seen, and let them provide the similarities for MAFA.

**Results** are given in Tab. 2 for both CLIP-ViT-B/32 and SigLIP-ViT-B/16. Surprisingly, we find that both GRIT and MAFA struggle in providing improvements to the classical CLIP recipe, sometimes even underperforming the IID baseline. Conversely, they consistently improve on the SigLIP recipe. This aligns with both ours and other recent observations (*e.g.*, Evans et al. (2024b); Udandarao et al. (2024)) that SigLIP models tend to benefit more from active batch sampling in general. However, despite this improvement, they lag far behind CABS-DM, which largely compares favorably. Even with SigLIP-ViT-B/16, CABS-DM improvements are as large of $+6.8\%$ on ImageNet and $+4.5\%$ when averaged across the entire zero-shot suite.

**Observations.** We speculate that the effectiveness of GRIT and MAFA for early VLMs does not transfer to modern LIP for a variety of reasons. For example, current LIP employs much larger batch sizes than early iterations (*e.g.*, 4096 vs 128 or 256), which potentially smoothens the impact of hard-negative sampling, simply due to the unlikelihood of having such large clusters of hard-negatives. Additionally, early VLMs were pretrained on much smaller datasets, thereby entering a multi-epoch regime and progressively refining the similarities computed for the *same samples* multiple times. Current LIP, however, relies on massive pretraining datasets that yield a *semi*-infinite data stream, with multi-epoch setups being far less common. In other words, current LIP is typically *compute-constrained*. In contrast, *data-constrained* pretraining regimes become necessary when the size of such massive datasets is largely reduced due to offline data filtering according to some criterion.

Table 2: **CABS-DM outperforms SOTA open-source batch sampling methods.** With both CLIP-ViT-B/32 (top) and SigLIP-ViT-B/16, CABS-DM provides significant benefits to LIP compared to Byun et al. (2022; 2024), making it a more suitable alternative to modern LIP.

| Method | Zero-shot Classification | | | | Retrieval | | Let-It-Wag! | Avg (Clf) |
|---|---|---|---|---|---|---|---|---|
| | IN-Val | IN-Shift | Obj | Scene | COCO | Flickr | | |
| ViT-B-32-CLIP | | | | | | | | |
| IID | 17.3 | 15.2 | 32.3 | 36.4 | 9.7 | **16.2** | 5.1 | 28.2 |
| GRIT-VLP | 17.6 | 15.0 | 31.7 | 35.6 | 9.7 | 15.6 | 6.3 | 27.5 |
| MAFA | 17.0 | 15.0 | 32.2 | 35.9 | 9.6 | 15.5 | 5.6 | 27.9 |
| CABS-DM | **21.9** | **18.6** | **34.2** | **37.0** | 7.7 | 12.6 | **7.5** | **31.0** |
| ViT-B-16-SigLIP-256 | | | | | | | | |
| IID | 17.2 | 15.3 | 29.6 | 35.9 | 11.1 | 18.9 | 5.2 | 26.4 |
| GRIT-VLP | 17.3 | 15.1 | 30.7 | 37.3 | **11.6** | **19.6** | 5.0 | 27.2 |
| MAFA | 17.2 | 15.2 | 30.7 | 36.2 | 10.5 | 19.4 | 5.2 | 27.1 |
| CABS-DM | **24.1** | **20.8** | **33.5** | **39.6** | 10.1 | 14.1 | **7.0** | **30.9** |

Table 3: **CABS-DM is compatible with CLIPScore filtering.** Although CABS-DM leads to more repeats, which yield diminishing returns on already curated data (Goyal et al., 2024), CABS-DM improves over IID even with $2\times$ more repeats across learning recipes.

| Method | Zero-shot Classification | | | | Retrieval | | Let-it-Wag! | Avg (Clf) |
|---|---|---|---|---|---|---|---|---|
| | IN-Val | IN-shift | Obj | Scene | COCO | Flickr | | |
| ViT-B-32-CLIP | | | | | | | | |
| IID | 27.3 | 23.0 | 39.8 | 43.1 | 13.8 | **24.1** | 10.7 | 35.7 |
| CABS-DM | **30.1** | **25.6** | **41.8** | **44.8** | **14.0** | 21.7 | **12.7** | **37.8** |
| ViT-B-16-SigLIP-256 | | | | | | | | |
| IID | 34.7 | 29.5 | 46.2 | **48.9** | 18.7 | **34.8** | 11.9 | 42.0 |
| CABS-DM | **37.5** | **32.2** | 46.2 | 48.5 | **18.9** | 29.3 | **12.6** | **42.7** |

Next, we analyze the properties of CABS-DM when training is *compute-* or *data-constrained*.

### 4.4 How does CABS-DM behave in data- vs compute-constrained regimes?

Modern pretraining ingredients, such as massive pretraining datasets, led to a critical distinction we aim to study: how does CABS-DM behave when pretraining is *data- vs compute-constrained*?

**Definitions.** In a nutshell, these two different settings can be defined with the following simple formalism. Let $C$ be the target compute for training (*i.e.* target FLOPs), $\mathcal{D}$ be the pretraining dataset, and $C_{\mathcal{D}}$ the compute spent for a full pass on $\mathcal{D}$ (*i.e.*, FLOPs per epoch). When $C \leq C_{\mathcal{D}}$, training is *compute-constrained*, which means compute is insufficient to train on all available data (or, at most, just as sufficient). Conversely, when $C > C_{\mathcal{D}}$, training is *data-constrained*, *i.e.*, data is insufficient w.r.t. the available amount of compute, rendering data repeats necessary.

**Experimental Design.** Because of CABS-DM's superbatches, Sec. 4 displays a data-constrained setting for CABS-DM and a compute-constrained setting for IID sampling: since a fraction $f{=}0.8$ of samples are filtered online, the effective data-per-epoch for CABS-DM is $5\times$ less than for IID, which instead operates with $C{=}C_{\mathcal{D}}$. To dissect this dichotomy further, we design two real-world experiments: ① *less data, but higher quality*, where we stick to the 128M sample budget, but reduce the 128M samples in DATACONCEPT via CLIPScore filtering, only keeping the top 30% and ending up with $\approx$38M samples.[2] To control for worst-case repeats, here we slightly decrease $f{=}0.5$; ② *long training*, where we match the DataComp-*large* scale with a budget of 1.28B samples seen.

**Less data, but higher quality.** The results are given in Tab. 3. Notably, CABS-DM shows full compatibility with CLIPScore filtered data, even though a worst-case scenario entails 6.67 repeats for CABS-DM, and only 3.33 for IID. Importantly, while repeating over already curated data is known to yield diminishing returns (Goyal et al., 2024), CABS-DM still trumps IID sampling.

---

[2]We use OpenAI's CLIP ViT-L/14 model for scoring.

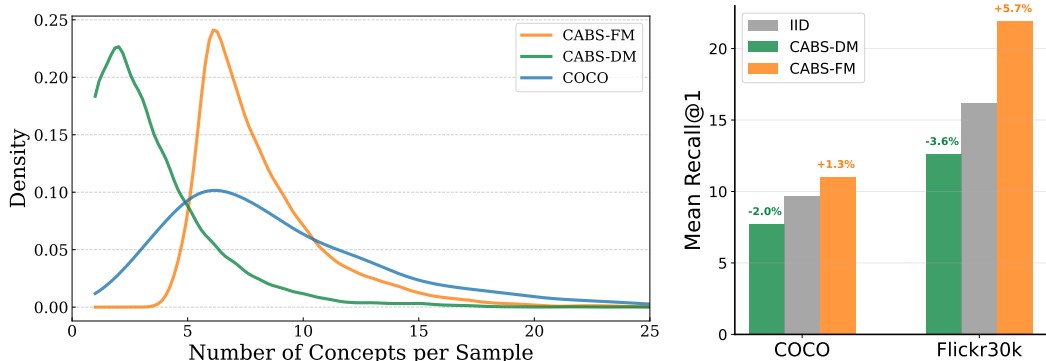

Figure 5: (*left*) KDE over concepts distributions yielded by CABS customizations (FM vs DM) and their alignment with COCO. (*right*) Corresponding performance on Retrieval benchmarks.

**Long Training** behavior along the training trajectory is portrayed by Fig. 4. Due to the large amount of compute required for this experiment, we only report CLIP-ViT-B/32, since its lower count of visual patches yields higher throughput than SigLIP ViT-B/16. From the figure, we can clearly see that, as long as training is compute-constrained for IID (dashed gray line), CABS-DM significantly outperforms the vanilla CLIP recipe, displaying an impressive $3.2\times$ data efficiency. Only after training has progressed far into the data-constrained regime, with CABS-DM yielding a worst-case of 50 repeats and IID yielding only 10, do the performance curves overlap. In a nutshell, these experiments confirm CABS-DM is fully compatible with real-world

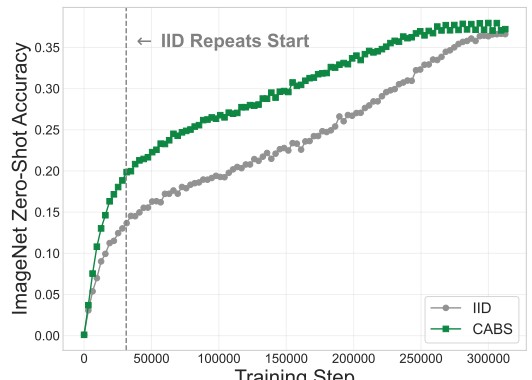

Figure 4: **Longer training** for CLIP ViT-B/32.

use cases of massive web-crawled data collections, or much smaller, highly curated resources.

## 5 OPTIMISING CABS FOR RETRIEVAL

As observed in Sec. 4.2, while the Diversity Maximisation variant of CABS delivers consistent improvements on zero-shot classification benchmarks, it underperforms on image–text retrieval tasks such as MSCOCO and Flickr30k. Why would a batch sampling strategy that consistently boosts classification performance not transfer to retrieval? We hypothesise that this gap stems from concept-based dataset compositions.

Classification datasets are typically dominated by single-object images, whereas retrieval datasets *often* feature multiple objects within a single sample and correspondingly richer scene compositions. To test this, we collect 4,096 random samples from MSCOCO and visualise the distribution of per-sample object counts, following the same protocol used to construct DATACONCEPT. As shown in Fig. 5, CABS-DM produces batches skewed towards single-object images, while MSCOCO exhibits a distribution skewed towards multi-object scenes.

To address this mismatch, we introduce a complementary CABS sampling algorithm that explicitly accounts for object multiplicity. Specifically, we rank all candidate samples by detected object count and filter away 80% of the super-batch (using a filter ratio $f = 0.8$). We refer to this variant as CABS-FM (Frequency Maximisation). Visualising the distribution of CABS-FM shows a sharper peak than MSCOCO, but one that overlaps with the same modal region, aligning more closely with the multi-object nature of retrieval datasets.

Does this alignment translate to downstream performance? Training a ViT-B-32 CLIP model with CABS-FM yields consistent improvements on both MSCOCO and Flickr30k. Being able to flexibly

adapt batch distributions at the pretraining stage, in ways that reflect the structural properties of target benchmarks, may thus provide a practical path towards task-aware optimisation.

With gains on retrieval benchmarks using CABS-FM, we posit that classification and retrieval evaluations favor divergent inductive biases during training batch construction. This observation resonates with recent work (Abbas et al., 2024b), which calls for separate curation strategies for classification and retrieval. To the best of our knowledge, CABS-DM and CABS-FM together represent the first public and reproducible demonstration of task-aware online batch sampling.

## 6 RELATED WORK

**Sampling Approaches for Training Foundation Models.** Training web-scale foundation models typically uses uniform, IID mini-batch sampling, which assigns equal weights to each data samples. However, in multimodal corpora, examples differ drastically in quality (Gadre et al., 2023; Schuhmann et al., 2022; Xu et al., 2023), are possibly redundant (Abbas et al., 2023; Elazar et al., 2023; Abbas et al., 2024a; Sorscher et al., 2022; Webster et al., 2023), and exhibit skewed, long-tailed distributions across concepts (Udandarao et al., 2024; Parashar et al., 2024). Moreover, for contrastive objectives like CLIP (Radford et al., 2021), batch composition heavily shapes the learning process. In this context, uniform sampling is not neutral: it can overexpose trivial or spurious correlations and under-represent rare but informative cases. Hence, several recent approaches try to apply better batch sampling schemes to ensure more effective cross-modal learning. Early approaches like RHO-Loss (Mindermann et al., 2022) and Bad-Students (Evans et al., 2024c) underscore the importance of more effective mini-batch sampling, but they only select data samples independently, without considering the overall batch composition. Works such as GRIT-VLP (Byun et al., 2022), MAFA (Byun et al., 2024), JEST (Evans et al., 2024a), B3 (Thirukovalluru et al., 2025), Falcon (Kim et al., 2025) and ACID (Udandarao et al., 2025) propose improved batch-aware sampling methods for optimizing the batch composition during each training step. Our paper builds on this line of works by incorporating concept diversity into training batch construction, an aspect missing from previous multimodal batch-sampling methods.

**Analyzing Concepts in Multimodal Datasets.** Understanding the composition of multimodal datasets is important for building better batch sampling methods. Early image-text datasets like CC-3M (Sharma et al., 2018), CC-12M (Changpinyo et al., 2021) and YFCC-100M (Thomee et al., 2016) partially characterize their inherent concept distributions using metadata from the original web sources where images are scraped from. The WebLI (Chen et al., 2022) dataset (used for training models like PaliGemma and SigLIP) was annotated using OCR models to detect objects in images. However, due to the large volume and compute resources required for annotating recent open datasets like LAION-5B (Schuhmann et al., 2022) and DataComp-1B (Gadre et al., 2023), very few works have studied their distribution of concepts. Udandarao et al. (2024) annotated the LAION-400M dataset by tagging each data sample with its constituent concepts by using a pretrained image-tagging model (Huang et al., 2023) and text search. Other works have proposed improving concept coverage in various ways, e.g. considering multilingual data Nguyen et al. (2024) or strategies for recaptioning Li et al. (2024). Our DATACONCEPT is also created through a curated pipeline that enriches samples with fine-grained concept annotations. However, a unique aspect of our work is that DATACONCEPT is specifically designed to enable explicit control over online, concept-based batch creation.

## 7 CONCLUSION

In this work, we investigate the role of incorporating concept-level information during large-scale language–image pretraining. In contrast to prior LIP dataset curation approaches that ignore concept-level information, we demonstrate that concept-awareness constitutes an underexplored yet crucial aspect. To this end, we introduced DATACONCEPT, a large-scale, fully annotated pretraining dataset designed to expose concept-level information, and CABS, a flexible framework for online, concept-aware batch sampling for LIP. Our extensive experimental evaluation demonstrates the benefits of CABS over IID and other batch sampling methods. Our results also reveal that CABS lead to improved performance across both classification and image–text retrieval tasks, demonstrating its versatility. By releasing DATACONCEPT and CABS as open resources, we aim to advance research on more robust and generalizable VLMs.

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

# Part I

# Appendix

## Table of Contents

# A  DATACONCEPT CURATION: FURTHER DETAILS

## A.1  VOCABULARY CONSTRUCTION

We scale up the tag generation pipeline of RAM++(Recognise Anything) (Huang et al., 2023) by incorporating more long-tailed concepts. In the original work, RAM++ extracts the top $4,585$ concepts by parsing 14 million sentences from their pool of pretraining datasets and then extracting tags using a SceneGraph Parser (Wu et al., 2019), hence attempting to focus on more common concepts.

However, our work focusses more on open-vocabulary recognition and localisation, hence we scale up the concept vocabulary to include objects that may be found in-the-wild in image-text pretraining datasets. We also adopt and filter the vocabulary pool from V3Det (Wang et al., 2023), a state-of-the-art open-vocabulary dataset which observes and encodes the relationship between categories by defining a hierarchy tree of concepts.

Curating this concept pool comes with redundancies, which need to be systematically resolved. Firstly, we perform a normalisation step to remove morphological variants of the same concept (*e.g* singular and plural forms) into a single entity using lemmatisation. Next, we remove semantic redundancies using WordNet(formalised through synsets) to detect synonyms. We then identify spelling/spacing artefacts and remove them if they are duplicated (`" cat"` and the correct `"cat"`). Finally, we identify unsafe concepts(*e.g.* racially motivated concepts like `white man` and `black man`) through thorough manual inspection and remove them.

## A.2 OBJECT TAGGING

Previous attempts to annotate pretraining datasets have used object tagging to return a list of probable objects in a sample ((Udandarao et al., 2024) used RAM++ Huang et al. (2023) to annotate visual concepts in many large image-text datasets. However, as discussed in Sec. 2, the expanded vocabulary introduces miscalibrations and overestimations in the model predictions. For example, abiding by the confidence threshold of $0.7$ image resolution of (384,384) from (Udandarao et al., 2024), we note that simply generating concept tags can lead to mistakes, a stricter regime is required. As a sanity check, we increase the confidence threshold to $0.75$ and still see miscalibrations in some form (see Fig. 6). Additionally, concept tags injects only one form of added metadata - other tasks like object detection can add richer and more valuable fine-grained information into these large datasets.

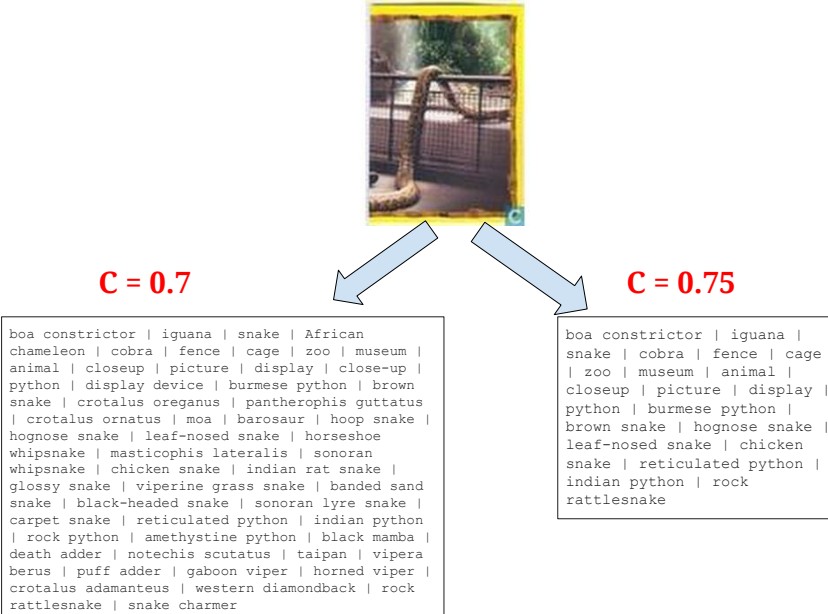

**C = 0.7**

```
boa constrictor | iguana | snake | African
chameleon | cobra | fence | cage | zoo | museum |
animal | closeup | picture | display | close-up |
python | display device | burmese python | brown
snake | crotalus oreganus | pantherophis guttatus
| crotalus ornatus | moa | barosaur | hoop snake |
hognose snake | leaf-nosed snake | horseshoe
whipsnake | masticophis lateralis | sonoran
whipsnake | chicken snake | indian rat snake |
glossy snake | viperine grass snake | banded sand
snake | black-headed snake | sonoran lyre snake |
carpet snake | reticulated python | indian python
| rock python | amethystine python | black mamba |
death adder | notechis scutatus | taipan | vipera
berus | puff adder | gaboon viper | horned viper |
crotalus adamanteus | western diamondback | rock
rattlesnake | snake charmer
```

**C = 0.75**

```
boa constrictor | iguana |
snake | cobra | fence | cage
| zoo | museum | animal |
closeup | picture | display |
python | burmese python |
brown snake | hognose snake |
leaf-nosed snake | chicken
snake | reticulated python |
indian python | rock
rattlesnake
```

Figure 6: RAM++ tends to overestimate classes when the vocabulary is expanded, even at high confidence thresholds. This arises from the increased semantic similarity among real-world concepts in the visual space, as a factor of a large vocabulary. This leads to an increase in the hierarchy for common and long-tailed classes(there are several sub-species of snakes in the vocabulary) and to some inherent uncertainty of making predicting for images that induce visual uncertainty.

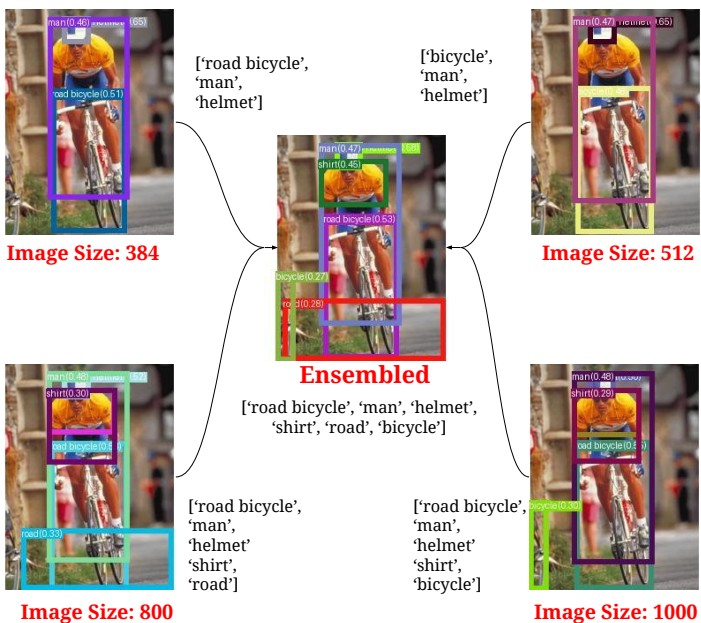

Figure 7: Using Weighted Box Fusion, we are able to detect single instances(no overlap of bounding boxes) of all relevant objects in an image.

## A.3 OBJECT DETECTION

Object tagging using RAM++ provides great insights into the object composition of images in inage-text datasets. However, relevant factors for the holistic understanding of pretraining data such as the number of instances of the same concept in an image(count) and the localisation of these concepts(spatial awareness) are confounded away by simply tagging an image with objects. To mitigate this, we indoctrinate bounding box information into the pipeline, which resolves both the issues identified.

Given an image and text pair, GroundingDINO returns bounding boxes, where each box is tagged with a similarity score across the individual input text tokens. GroundingDINO can effectively detect objects from an image when provided an input text. Some approaches involve having a user-defined text prompt and providing the entire pool of concepts as the text input. Both these approaches have their set of problems: firstly, we are unable to individually define prompts when dealing with pretraining data as it involves manually annotating millions of images, and secondly, providing the entire pool of concepts leads to over-representation of objects being detected which are not visually present in the image, thus leading to some form of hallucination.

Our solution involves providing RAM++ object tags at a $0.75$ confidence threshold as prompts to GroundingDINO. By reducing the vocabulary pool, we mitigate hallucinations and errors while also improving the detection model's processing speed. Through manual inspection, to remove low-confidence predictions to prevent a second degree of over-representation, we set a text threshold by only extracting concepts with a box-concept similarity score higher than $0.27$. We set the same threshold for bounding box confidence scores too. With this configuration, we can now annotate each image of a pretraining dataset with the concept tags, per-concept logit scores from RAM++ and the set of bounding boxes, detected classes and their corresponding confidence scores.

An additional confounder is that DataComp-128M is available in multiple resolutions. To leverage this and increase the trustworthiness of DATACONCEPT, we apply Weighted Box Fusion (WBF) Solovyev et al. (2021) for bounding box ensembling. WBF generates the final set of bounding boxes by using the confidence scores of the proposed bounding boxes of multiple object detection models/various configurations of the same object detection model. This approach is in contrast to Non-maximum suppression(NMS) which just removes part of the predictions instead of aggregating them. Ensem-

bling has proven to be an effective strategy in complex object detection tasks (Tuggener et al., 2024). Specifically, we ensemble across image resolutions $384, 512, 800, 1000$ to obtain more robust final detection results, refer to Fig. 7 for visual inspection.

As we have demonstrated, DATACONCEPT has been curated using high confidence thresholds and stricter annotation protocols, with localisation requiring bounding boxes to be generated for the precise regions of objects. This added difficulty has led to extremely rare concepts being underrepresented in the annotations. Nevertheless, DATACONCEPT-M contains $6,201$ unique concepts, which we define as $\mathcal{C}$, the concept pool for CABS. How these concepts are distributed and how concept-dense they are can be found in Fig. 8.

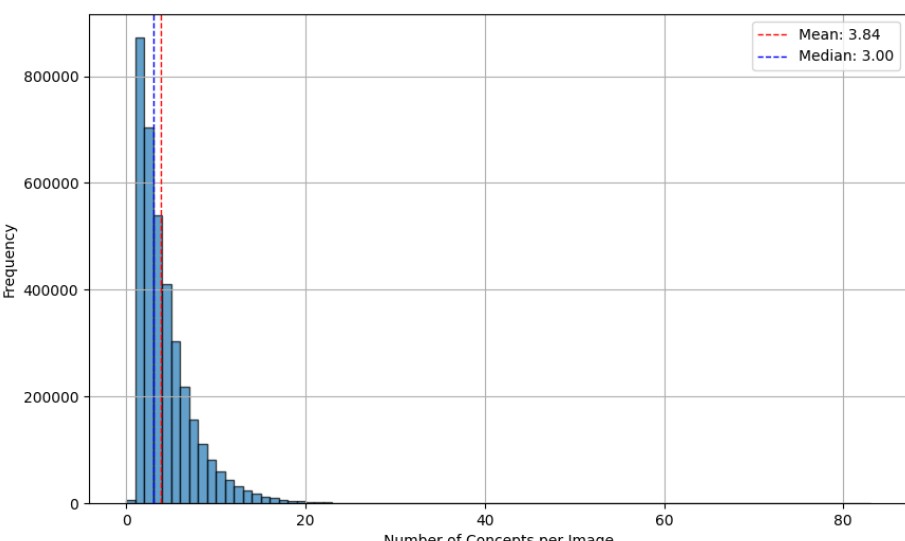

Figure 8: **What is the representation of concepts in web-scale pretraining samples?** We demonstrate the distribution of concept counts per sample after annotations using GroundingDINO, using optimised thresholding and confidence scores. that GroundingDINO can predict a concept many times, hence these numbers reflect the total number of concepts detected in an image, not unique concepts.

## B EXTENDED BENCHMARK PERFORMANCE: MORE MODELS

Table 4: **Extended Results** including CLIP ViT-S-16 and SigLIP ViT-SO400M. CABS-DM delivers consistent improvements with these variants as well.

| Method | Captions | Zero-shot Classification | | | | Retrieval | | Let-it-Wag! | Avg (Clf) |
|---|---|---|---|---|---|---|---|---|---|
| | | IN-Val | IN-shift | Obj | Scene | COCO | Flickr | | |
| *ViT-S-16* | | | | | | | | | |
| IID | alt | 16.9 | 15.0 | 30.3 | 35.4 | 9.6 | **17.4** | 6.1 | 26.6 |
| CABS-DM | alt | **24.6** | **20.6** | **34.8** | **39.0** | **10.3** | 16.1 | **8.3** | **31.5** |
| IID | recap | 24.8 | 22.8 | 39.4 | 44.4 | **28.7** | **47.2** | 6.3 | 35.4 |
| CABS-DM | recap | **30.0** | **27.4** | **40.6** | **45.0** | 27.5 | 41.7 | **8.0** | **37.8** |
| *ViT-B-32* | | | | | | | | | |
| IID | alt | 17.3 | 15.2 | 32.3 | 36.4 | **9.7** | **16.2** | 5.1 | 28.2 |
| CABS-DM | alt | **21.9** | **18.6** | **34.2** | **37.0** | 7.7 | 12.6 | **7.5** | **31.0** |
| IID | recap | 21.7 | 20.8 | 36.4 | 43.1 | **24.0** | **41.3** | 5.9 | 33.0 |
| CABS-DM | recap | **26.7** | **25.4** | **39.6** | **42.8** | 22.8 | 34.4 | **7.1** | **35.5** |
| *ViT-B-16-SigLIP-256* | | | | | | | | | |
| IID | alt | 17.2 | 15.3 | 29.6 | 35.9 | **11.1** | **18.9** | 5.2 | 26.4 |
| CABS-DM | alt | **24.1** | **20.8** | **33.5** | **39.6** | 10.1 | 14.1 | **7.0** | **30.9** |
| IID | recap | 28.8 | 27.4 | 41.5 | 48.9 | **37.1** | **57.0** | 6.6 | 38.6 |
| CABS-DM | recap | **34.7** | **32.3** | **43.2** | **50.6** | 36.0 | 51.4 | **7.6** | **41.1** |
| *ViT-SO400M-14-SigLIP* | | | | | | | | | |
| IID | alt | 15.5 | 13.7 | 27.5 | 34.7 | 8.8 | 13.7 | 4.7 | 24.5 |
| CABS | alt | **22.6** | **18.8** | **33.4** | **40.0** | **11.3** | **15.9** | **6.2** | **30.2** |
| IID | recap | 34.1 | 31.8 | 46.3 | 55.9 | 37.7 | **53.8** | 7.6 | 42.2 |
| CABS | recap | **39.6** | **36.1** | 45.1 | **57.5** | **39.0** | 52.3 | **9.4** | **44.2** |

To provide a more in-depth analysis of the trends seen when comparing IID sampling and CABS-DM, we conduct experiments on two additional models, CLIP ViT-S-16 and SigLIP ViT-SO400M. We arrive at the same conclusions as discussed in Sec. 4.2, we see strong performance boosts on classification datasets with a dip in performance on retrieval datasets. However, we see some outliers such as ViT-S-16 on MSCOCO for alt-text (+0.7%) as well as for the ViT-SO400M-14-SigLIP models. However, the perdomance gains aren't universal or significant enough to conclude that CABS-DM is an ideal sampling algorithm for retrieval. We defer to CABS-FM for retrieval-optimised batch sampling.

# C CONTINUAL PRETRAINING

Table 5: **Continual Pretraining Experiments.** In line with the evidence of Sec. 4, CABS improves over IID sampling even when continually pretraining an IID checkpoint.

| Method | Captions | Zero-shot Classification | | | | Retrieval | | Let-it-Wag! | Avg (Clf) |
|---|---|---|---|---|---|---|---|---|---|
| | | IN-Val | IN-shift | Obj | Scene | COCO | Flickr | | |
| ViT-B-32 | | | | | | | | | |
| IID | alt | 23.7 | 20.0 | 37.7 | 42.3 | **13.7** | **24.5** | 7.9 | 33.4 |
| CABS-DM | alt | **27.8** | **23.9** | 37.4 | **42.7** | 10.7 | 16.9 | **8.9** | **34.4** |
| iid | recap | 27.7 | 25.8 | 41.7 | 47.7 | **30.5** | **49.0** | 7.7 | 38.1 |
| CABS-DM | recap | **31.7** | **29.1** | **43.4** | 46.8 | 29.7 | 43.7 | **8.9** | **40.0** |

We wish to see ifCABS-DM is a strong batch sampling algorithm on other pretraining regimes as well. To this end, we adopt a continual pretraining paradigm, where checkpoints trained at the same scale (128M samples seen) are used as model weights at the beginning of a new training run. In our case, we use a CLIP ViT-B-32 checkpoint, trained on DataComp-128M and compare IID-sampling and CABS-DM. We still see CABS-DM outperforming IID sampling in almost all settings, across benchmarks and text distributions (alt-text and concept-aware synthetic recaptions).

# D    ABLATION ON FILTER RATIOS

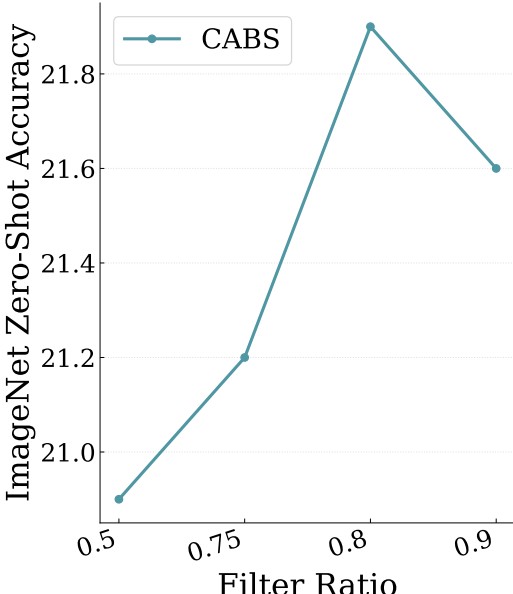

Figure 9: **CABS-DM is affected by varying batch filter-ratio**.

In this section, we show how the filter ratio $f$, defined as the parameter that determines the size of a sub-batch $b$ given super-batch of size $B$. For example, a filter ratio of $f = 0.5$ would correspond to a super-batch of size $8192$ for a sub-batch of size $4096$. In most of our experiments, we fix the filter ratio to $0.8$. Fig. 9 provides an ablation over various other filter ratios for a ViT-B/32 CLIP model, tested on ImageNet across filter ratios $0.5, 0.75, 0.8, 0.9$. Performance trends over the set of filter ratios indicate that $0.8$ is indeed the optimal filter ratio at the 128M sample scale.

# E  FINE-GRAINED BENCHMARK PERFORMANCE

Figure 10: Dataset-wise comparisons for all benchmarks of Sec. 4 for CLIP ViT-B/32 between CABS-DM and IID sampling. A positive performance difference indicates a benchmark where CABS-DM outperforms IID sampling.

We provide an expanded probe into the specific benchmarks where CABS-DM shows performance boosts over IID-sampling in 25 out of 28 benchmarks. With this, we can ascertain that despite maximising for concept diversity, CABS-DM shows strong gains on datasets that test for long-tailed concepts as well as for more common concepts. This confirms that CABS-DM is an all-round performant batch sampling algorithm for classification tasks.

