# OpenReview forum: "Concept-Aware Batch Sampling Improves Language-Image Pretraining"
_ICLR.cc/2026/Conference — ICLR 2026 Conference Withdrawn Submission_

### Official Review · Reviewer_nu4g · 2025-10-27

**Soundness:** 2
**Presentation:** 2
**Contribution:** 2
**Rating:** 4
**Confidence:** 5

**Summary:**

The paper proposes: (1) a pipeline that automatically generates fine-grained, concept-focused image captions using state-of-the-art tagging, grounding, and VLM models; and (2) evidence that pretraining image-language model with batches whose concepts are uniformly distributed outperforms pretraining the same model using concept-biased batches. Combining these contributions yields considerable accuracy gains across diverse image–language pretraining settings, including different loss functions and base models.

**Strengths:**

The paper shows that (1) relabeling captions to emphasize image concepts (DataConcept) and (2) pretraining with concept-balanced batches (CAPS-DM) consistently improve zero-shot classification across diverse settings. Zero-shot retrieval accuracy also increases with the optimal batching strategy (CAPS-FM). These conclusions are supported by extensive experiments.

**Weaknesses:**

- The method performs well on zero-shot classification but degrades zero-shot retrieval accuracy under the default CAPS-DM. Although the CAPS-FM variant improves retrieval, relying on different setups for different applications weakens the contribution, as a single pretrained model is generally expected to work across tasks. If the model underperforms on either classification or retrieval, it may also struggle on downstream tasks such as detection and segmentation, which undermines the promise of foundation models.

- Developing a data relabeling pipeline for pretraining are already well studied (e.g., https://arxiv.org/pdf/2311.06242 and references therein). To establish novelty, more comprehensive qualitative and quantitative comparisons are needed; comparisons limited to naive baselines (IID) and a hard-negative mining variants are insufficient.

**Questions:**

- If CAPS-FM is applied for zero-shot classification, is that better than the baseline IID? I.E. Can the CAPS-FM be more general than CAPS-DM so that CAPS-FM works in both the applications (zero-shot classification and retrieval)?

- Have author tried CAPS with the existing dataset without using DataConcept? This is important to understand what is the major contribution of the improvement. Is it because of DataConcept or CAPS?

- What is the baseline performance on the benchmark even without DataConcept?

- The ImageNet zeroshot accuracy in Figure 1 is generally too low. If author is using variant of ImageNet zeroshot benchmark, please clearly specify them in detail in the Figure 1.

---

### Official Review · Reviewer_fQRo · 2025-10-30

**Soundness:** 2
**Presentation:** 3
**Contribution:** 2
**Rating:** 4
**Confidence:** 3

**Summary:**

This paper studies the role of incorporating concept-level information during large-scale language–image pretraining, which is underexplored in the literature. To this end, the paper introduces DATACONCEPT, a large-scale, fully annotated pretraining dataset, which augments samples with fine-grained concept annotations and concept-driven synthetic captions. Besides, this paper proposes a flexible framework, CABS, for online, concept-aware batch sampling for LIP. Empirical experiments demonstrate the benefits of CABS over IID and other two batch-sampling baselines.

**Strengths:**

1.	The constructed concept-aware pretraining dataset with rewritten context-aware captions seems promising and useful.

2.	The experiment in table1 shows a clear advantage of concept-aware re-captions in LIP.

**Weaknesses:**

1.	While the curated concept-aware pretraining dataset is meaningful to the community, it seems the proposed CABS doesn’t work well with the dataset. Different from other sampling strategies that may achieve consistent improvement across classification and retrieval tasks, CABS may perform well on classification but worse on retrieval tasks. Though the authors proposed an alternative CABS-FM to improve performance on retrieval tasks, it makes the total design complicated since a hard choice needs to be made and the trained model can't perform different tasks effectively, which means that it loses the advantage of pretrained models to generalize well across tasks..

**Questions:**

1.	Although the authors claim that Evans et al. (2024a) and Udandarao et al. (2025) didn’t release their code, is it possible to reproduce them since their methods show strong performance and the paper only has two baselines?

2.	It’s better to include an algorithm (pseudo code) to show how CABS-DM processes samples during training. Only natural language description makes it difficult to follow and understand the procedure clearly.

3.	What are the guidelines for choosing the hyperparameter f? Is it sensitive?

4.	Please explain more about $f_c$.

---

### Official Review · Reviewer_MhEs · 2025-10-31

**Soundness:** 3
**Presentation:** 3
**Contribution:** 1
**Rating:** 2
**Confidence:** 5

**Summary:**

The paper annotates 128M image-text pairs from DataComp XLarge, and proposes Concept-Aware Batch Sampling (CABS), a diversity-driven strategy that e.g. samples mini-batches based on the diversity of their constituent concepts instead of random sampling. By balancing concept coherence and intra-batch diversity, CABS yields improved performance across vision and language tasks.

**Strengths:**

- Achieves better zero-shot performance than random sampling. Another heuristic variant that samples examples with more concepts yields a better performance for retrieval.

- Provides empirical validation on benchmarks.

**Weaknesses:**

I don't see the novelty or new insights provided by this paper. The idea that balanced mini-batches improve the convergence and performance on smaller groups of data is well-known. This idea has been used before in many different domains, including federated learning, data selection, etc. From optimization perspective, the reason is that balanced mini-batches have smaller gradient variance which yield faster convergence, which is theoretically analyzed and shown by several existing papers in the literature (this is a relatively old concept). Besides, upsampling underrepresented groups is obviously beneficial. The main idea of the paper is to annotate the concepts in training example and use them to sample balanced mini-batches. While this is a good usage in a production pipeline, I don't see any new "scientific" insight. If the main contribution is the annotations, the paper is more suitable for the dataset and benchmark track.

**Questions:**

What's the new scientific insight (finding, method, analysis, etc) from this paper, in authors' opinion?

---

### Official Review · Reviewer_Vhed · 2025-11-01

**Soundness:** 3
**Presentation:** 4
**Contribution:** 3
**Rating:** 6
**Confidence:** 5

**Summary:**

This paper advocates for concept-aware data curation for contrastive language-image pre-training (CLIP) and first proposes a dataset DataConcept based on a subset of DataComp, and consists of 128M image-text pairs, where each image annotated with fine-grained concepts, bounding boxes, and concept-aware synthetic captions; Then, the method propose a Concept-Aware Batch Sampling strategy to improve the training effectiveness. Different from MetaCLIP, it seems that this method do not need to build a balanced dataset offline but adaptively adjust the sampling throughout the training to maintain a balanced distribution seen by the model. The experimental evaluation shows untrivial gain on classification and long-tailed benchmarks.

**Strengths:**

- A new data curation mechanism with online method to adjust the data distribution seen by the model to improve the training effectiveness.
- The motivation to change existing random sampling is appreciated.
- The evaluation is comprehensive.

**Weaknesses:**

- The definition of concept is critical for the method development. Then, the discussion on why the current concept bank definition is optimal is needed. As the author mentioned MetaCLIP many times, I am curious how the concept bank different from the metadata used in MetaCLIP (in MetaCLIP, the balanced distribution according to metadata is one critical standard for MetaCLIP dataset construction.
- Following, when the concepts bank contains erroneous or missed concepts, how your method can robustly expand or update it in an online manner.
- Performance trade-off: By comparing performance in Table 1 and Fig. 5, the CABS-DM helps classification but hurts retrieval, while the CABS-FM favors retrieval only. Then, I am curious whether these two mechanism variants can be combined, or whether it is always conflicting for optimizing the performane for classification & retrieval.
- For your method, I wanna check whether the image encoder during training must be frozen or can also be updated.

**Questions:**

For multi-modal pre-training, the current method primarily uses text to determine the context in batch sampling, which is similar to [2]. However, whether the visual information can also be utilized (e.g., the team of MetaCLIP also propose CIT for visual pre-training).

[1 ]CIT: Curation in training for effective vision-language data (ICCV)

[2] In-context pretraining: Language modeling beyond document boundaries (ICLR)

Please see my comments in the weakness. Happy to increase the score if all of my questions & weakness can be properly addressed.

---

### Author Response · Authors · 2025-11-14

We thank the reviewers for their time and intend on addressing the points made in a systematic manner:

1. how the concept bank different from the metadata used in MetaCLIP(Vhed): Our concept curation is independent of MetaCLIP curation. They use 500k queries from Wikipedia (“base query list is all words occurring at least 100 times in the English version of Wikipedia, augmented with bi-grams with high pointwise mutual information as well as the names of all Wikipedia articles above a certain search volume. Finally all WordNet synsets not already in the query list are added”). Due to this, prepositions, articles and other part-of-speech variants other than nouns exist in the pool. Additionally, it only considers single-word queries. Our concept pool is different as it only considers nouns that can be visually localised, as we collect concepts from the vocabularies of works focussing on computer vision tasks(classification, object detection, referring segmentation, etc).

2. What do you do when the concepts bank contains erroneous or missed concepts: a concept bank of 19261 nouns is fairly comprehensive, however we acknowledge that concepts may be missing. MetaCLIP suffers from the same problem - the metadata has one more missing concept whenever a new Wikipedia article is created

3. “paper only has two baselines”: we intend to scale up baseline comparisons by including MetaCLIP. ACID and JEST remain close-sourced rendering a faithful reproduction difficult.

4.“better to include an algorithm (pseudo code)”(fQRo): thank you for the suggestion, we will do so.

5.“Please explain more about f_c”(fQRo): It is the  frequency of a concept(number of times it exists in a superbatch) from our concept bank  in superbatch before balancing

6.“hard choice needs to be made and the trained model can't perform different tasks effectively”: Recent works have shown that separate curation is indeed necessary [1](we have discussed this in the submission too). We additionally show proof that retrieval and classification is distributionally divergent by using the same pipeline to annotate COCO as we used to construct DataConcept for a fair comparison.

[1]https://www.datologyai.com/blog/productionized-multimodal-data-curation-at-the-billion-sample-scale


Minor/major oversights from reviewers:

1.“the paper is more suitable for the dataset and benchmark track.” Please note that we did indeed submit to the datasets and benchmark track

2.“No scientific insight”(Mhes): Three major insights include online batch sampling trumps offline curation, performance boosts by sample repeats contribute more to model performance than previously estimated, old methods(GRIT, MAFA) don’t scale to modern regimes.

3.“Current method primarily uses text to determine the context in batch sampling”(Vhed): we would like to clarify that our concept annotations are not text-based: they are distilled from visual information. This is fundamentally different from MetaCLIP, which relies solely on textual metadata.

4.“What is the baseline performance on the benchmark even without DataConcept?”(nu4g): That is equivalent to training on DataComp.
“The ImageNet zeroshot accuracy in Figure 1 is generally too low.” (nu4g): the results are in accordance to the number of samples seen (128M). Please refer to the Datacomp paper for a reference.

---

### Note · Authors · 2025-11-14

I have read and agree with the venue's withdrawal policy on behalf of myself and my co-authors.